# The Anti-Inflammatory Properties of Licorice (*Glycyrrhiza glabra*)-Derived Compounds in Intestinal Disorders

**DOI:** 10.3390/ijms23084121

**Published:** 2022-04-08

**Authors:** Camila dos Santos Leite, Gabriel Alves Bonafé, Juliana Carvalho Santos, Carlos Augusto Real Martinez, Manoela Marques Ortega, Marcelo Lima Ribeiro

**Affiliations:** 1Laboratory of Immunopharmacology and Molecular Biology, São Francisco University Medical School, Bragança Paulista 12916-900, Brazil; camilayantony@gmail.com; 2Laboratory of Cell and Molecular Tumor Biology and Bioactive Compounds, São Francisco University Medical School, Bragança Paulista 12916-900, Brazil; gabrielbonafe@outlook.com; 3Lymphoma Translational Group, Josep Carreras Leukaemia Research Institute (IJC), 08916 Badalona, Spain; j.carvalho@carrerasresearch.org; 4Department of Surgery and Proctology, São Francisco University (USF), Bragança Paulista 12916-900, Brazil; carlos.martinez@usf.edu.br

**Keywords:** *Glycyrrhiza glabra*-derived compounds, glycyrrhizin (G), glycyrrhetinic acid (GA), dipotassium glycyrrhizinate (DPG), inflammation, oxidative stress, intestinal disorders

## Abstract

Intestinal diseases, such as inflammatory bowel diseases (IBDs) and colorectal cancer (CRC), are a significant source of morbidity and mortality worldwide. Epidemiological data have shown that IBD patients are at an increased risk for the development of CRC. IBD-associated cancer develops against a background of chronic inflammation and oxidative stress, and their products contribute to cancer development and progression. Therefore, the discovery of novel drugs for the treatment of intestinal diseases is urgently needed. Licorice (*Glycyrrhiza glabra*) has been largely used for thousands of years in traditional Chinese medicine. Licorice and its derived compounds possess antiallergic, antibacterial, antiviral, anti-inflammatory, and antitumor effects. These pharmacological properties aid in the treatment of inflammatory diseases. In this review, we discuss the pharmacological potential of bioactive compounds derived from Licorice and addresses their anti-inflammatory and antioxidant properties. We also discuss how the mechanisms of action in these compounds can influence their effectiveness and lead to therapeutic effects on intestinal disorders.

## 1. Introduction

Licorice (*Glycyrrhiza glabra*) has been used in traditional Chinese medicine for thousands of years. Clinically, it is used widely to treat immune systems, respiratory, and digestive diseases [1,2,3,4,5,6], and no severe side effects have been reported so far [7]. In addition, Licorice-derived compounds possesses antiallergic, antibacterial, antiviral, anti-inflammatory, and anticarcinogenic effects [8,9,10]. These pharmacological properties aid in inflammatory disease treatment [11,12,13] (Figure 1).

The main bioactive compounds isolated from Licorice are glycyrrhizin (G) and glycyrrhetinic acid (GA) [14]. G is a triterpene glycoside complex and has been shown to possess cytotoxic effects against several cancer cell lines such as colon, lung, leukemia, melanoma, and glioblastoma (GBM) [9,15,16,17,18,19,20,21]. Additionally, the incidence of liver carcinogenesis in patients with hepatitis C was clinically reduced after G administration [22]. GA, an aglycone of G, has been demonstrated to have pro-apoptotic effects on human hepatoma, promyelocytic leukemia, stomach cancer, Kaposi sarcoma-associated herpesvirus-infected cells, and prostate cancer cells in vitro by inducing DNA fragmentation and oxidative stress [23,24,25]. In addition, several genotoxic studies have indicated that G is neither teratogenic nor mutagenic and may possess anti-genotoxic properties under certain conditions [26,27]. As a result, there is a high level of use of Licorice and GZ in the US with an estimated consumption of 0.027–3.6 mg/kg/day [27].

However, GA oral efficacy is impaired due to its low solubility and permeability through the gastrointestinal mucosa [28]. It has been shown that GA administered through nanocarriers (GA-F127/TPGS-MMs) [29], micellar carrier based on polyethylene glycol-derivatized GA (PEG-Fmoc-GA) [30], and microparticles [31] increase absorption significantly [28,29,30,31]. Both G and GA have been prescribed for several therapeutic purposes, such as cancer and inflammation; however, side effects have pointed out the problem of their toxicity [32].

Dipotassium glycyrrhizinate (DPG), a dipotassium salt of GA, has been recently used as a flavoring and skin conditioning agent with demonstrated anti-allergic and anti-inflammatory properties [32]. It can inhibit leukotriene and reduce histamine levels with an apparent lack of adverse side effects [32,33,34]. In addition, it has been demonstrated that DPG has anti-inflammatory, antioxidant, immunomodulatory, anti-ulcerative, and antitumoral properties [11,13,35].

In this context, this review examines recent studies on the pharmacological properties of some bioactive compounds derived from Licorice and addresses their anti-inflammatory and antioxidant properties, as well as their therapeutic effects on gastrointestinal disorders.

## 2. G, GA, and DPG-Mediated Anti-Inflammation Regulation

As stated previously, Licorice compounds such as G, GA, and DPG have anti-inflammatory, antioxidant, antiviral, immunomodulatory, and antitumor properties [11,12,13]. Inflammation is an evolutionarily conserved, tightly regulated protective mechanism that comprehends immune, vascular, and cellular biochemical reactions. The normal inflammatory response is temporally restricted and, in general, beneficial to the host. Chronic inflammatory response, on the other hand, is a risk factor for the development of several diseases such as ischemic heart disease, stroke, cancer, and diabetes mellitus, among others [36,37]. Taking this into account, natural compounds have been widely used to treat all sorts of inflammatory conditions.

The anti-inflammatory effects of G and GA have long been reported. G has exerted anti-inflammatory actions by inhibiting the generation of reactive oxygen species (ROS) by neutrophils, the most potent inflammatory mediator at the site of inflammation [38]. Moreover, G has enhanced interleukin (IL)-10 production by liver dendritic cells in mice with hepatitis [39]. GA has presented anti-inflammatory and anticarcinogen effects on several tumor cell lines such as human hepatoma (HLE), promyelocytic leukemia (HL-60), stomach cancer (KATO III), and prostate cancer (LNCaP e DU-145) by both DNA fragmentation and gene deregulation required for oxidative stress control [23,24,25].

More recently, one study exposed the U251 GBM cell line to different concentrations of GA (1 mM, 2 mM, and 4 mM), and the authors observed that the inhibition of cell proliferation and colony formation, apoptosis stimulation, and significant decrease in p65 protein, which is responsible for the activation of the nuclear factor kappa B (NF-κB) pathway [40]. The NF-κB pathway is constantly activated in GBM and is responsible for the aggressiveness of the disease and regulation of the expression of anti-apoptotic genes, and cell adhesion and invasion factors [41]. Thus, some studies have suggested that the inhibition of the NF-κB pathway could decrease the resistance of tumor cells to chemotherapy and contribute to increasing the survival of patients with GBM [42,43,44,45].

Recently, a study has shown that DPG exposure has anti-tumoral effects on GBM cell lines (U87MG and T98G) through cell proliferation decrease and apoptosis stimulation. Furthermore, DPG anti-tumoral effect was related to NF-κB pathway suppression by *IRAK2* and *TRAF6* mediating miR-16 and miR-146a, respectively. Finally, the authors have also shown that DPG was able to inhibit the subpopulation of stem cells essential for tumor formation, survival, and recurrence [46].

## 3. G, GA, and DPG-Mediated Crosstalk between Inflammation and Oxidative Stress Pathways

Oxidative stress consists of an imbalance of endogenous pro-oxidant and antioxidant activities, characterized by excessive formation of high ROS and reactive nitrogen species (RNS) [47]. Small amounts of ROS are synthesized physiologically and act on cell homeostasis; however, in the disease context, the excessive synthesis of ROS disrupts the antioxidant defense system, causing cellular apoptosis [47]. This condition is commonly associated with oxidative changes such as lipid peroxidation, protein carbonylation, carbonyl adduct, nitration, and DNA impairment as well as the induction of inflammatory processes, leading to several diseases [48,49]. Cyclooxygenase type 2 (Cox-2) and inducible nitric oxide synthase (iNOS) enzymes, responsible for the release of pro-inflammatory mediators, prostaglandin E2 (Pge-2), and nitric oxide (NO), play relevant roles in oxidative and acute inflammatory processes [49].

The high mobility group box 1 (Hmgb1) cytokine plays an important role in the pathologic process of endothelial permeability under oxidative stress [49]. DPG and G have presented antioxidant effects due to their negative modulation of Hmgb1 in the DSS-induced colitis mice model [49]. It has been shown that G inhibits Hmgb1-cytokine secretion by blocking the Cytochrome C release and caspase-3 activity, consequently inhibiting apoptosis in inflammation-related stroke rat models [50,51]. In addition, the G compound decreases the iNOS, TNF-α, IL-1β, and IL-6 expression levels by the modulation of p38 mitogen-activated protein kinases (p38-MAPK) and c-Jun *N*-terminal kinase (p-JNK) signaling pathways in brain vascular cells [51] and by preventing oxidative stress and apoptosis through the inhibition of p38-MAPK, p-JNK, and NF-κB signaling pathways in lung cells [52].

Accordingly, the G compound can inhibit oxidative stress and inflammatory response by attenuating the activity of the Hmgb1 and NF-κB signaling pathways, with decreased levels of malondialdehyde (MDA) and cytokines (TNF-α, IL-1β and IL-6) in lung cells [53]. Moreover, G increases glutathione-S-transferase (GSTs) levels, decreases MDA, and negatively regulates the expression of TNF-α, IL-6, iNOS, and monocyte chemotactic protein-1 (MCP-1) in liver cells [54]. G compound has been shown to suppress NF-κB pathway through inhibiting the toll-like receptor 4 (TLR4) in renal cells [55] and reducing the formation of intracellular ROS. Moreover, an activation of the AMP/nuclear factor erythroid-2-related factor-2 (NRF2) pathways in vitro was observed, positively regulating the antioxidant enzymes, HO-1, NQO-1, and GCLC and negatively regulating TNF-α, IL-1β, and IL-6 [56].

According to descriptions, GA also suppresses oxidative stress and neuroinflammation induced by A1C13 through TLR4/NF-κB signaling pathway inhibition [57]. In accordance, one study has observed that GA was able to attenuate oxidative stress and neuroinflammation induced by rotenone reducing the activation of the ionized calcium-binding adapter molecule-1 (Iba-1), preventing glutathione depletion, lipid peroxidation inhibition, and attenuation of the induction of COX-2 and iNOS [58]. In addition, a restored mitochondrial complex I and IV, a reduction in the generation of ROS, the release of Cytochrome C, and ultimately cell apoptosis inhibition after exposure to GA in brain tissue of adult Sprague Dawley Rats were observed [59].

GA can suppresses lipopolysaccharide (LPS)-induced oxidative stress, inflammation, and apoptosis through activation of the extracellular signal-regulated kinase (ERK) pathway, and inhibition of the NF-κB in renal cells [60]. GA also suppresses oxidative stress and inflammation through activation of the NRF-2 and HO-1 pathways and IκB and NF-κB p65 signaling inhibition in cardiac cells [61].

In the liver tissue of rats, GA inhibits NTiO2-induced apoptosis by superoxide dismutase (SOD) and glutathione peroxidase (GPx) activation [61]. Moreover, it has been shown that GA can inhibit caspase-3 and -9 at mitochondria in HepG2 cells, positively and negatively regulating Bcl-2 and Bax proteins expression, respectively [62]. Table 1 summarizes the studies used in this review.

## 4. Therapeutic Effect of G, GA, and DPG for Intestinal Disorders

Crohn’s Disease (CD) and Ulcerative Colitis (UC) are the main inflammatory bowel diseases (IBD) that affect the gastrointestinal tract. These diseases are characterized by chronic and progressive inflammation of the gastrointestinal tract, associated with extraintestinal manifestations such as arthritis, uveitis, erythema nodosum, gangrenous pyoderma, and cholangitis [74,75]. It has been shown that inflammatory reaction and the increased production of ROS are commonly associated with the pathogenesis of IBD [49,76]. ROS are toxic to cells and their overproduction causes breakage of the various lines of defense that make up the mucosal barrier [76]. The dysregulation of the immune response and the exaggerated release of pro-inflammatory/interleukin cytokines (IL-1, IL-6, IL-8, and TNF-α) culminate in the exacerbation of intestinal inflammation [63,64]. Thus, oxidative stress is considered an initial step to the colonic epithelium inflammation of patients with IBD.

Patients with IBD are at increased risk of developing colorectal cancer (CRC). Epidemiological data from patients with UC estimate that the risk of CRC is approximately 2- to 3-fold more than the general population, and patients with CD appear to have a similar increased risk [77]. Chronic inflammation is the most important aspect of neoplastic progression, resulting in dysplastic precursor lesions that may arise from different areas of the colon [78]. The overproduction of ROS can damage the DNA of the chronically inflamed colonic mucosa cells, increasing the mutation rate in genes related to development of CRC [76,79]. Oxidative reactions are an integral part of the inflammatory response and are generally associated with CRC development [80]. The potential mechanisms for the natural alkaloids in the treatment of UC has been recently described, showing that its positive effects are closely related to the modulation of oxidative stress, immune response, intestinal microbiota, and improvement of the gut barrier function [81]. Considering that oxidative stress is one of the main factors related to the development of IBD and IBD-related CRC, it is possible that the use of active principles found in *Glycyrrhiza glabra* extract may be effective for the treatment and prevention of both diseases [82] (Table 1).

Several studies have evaluated the effectiveness of G, GA, and GL in experimental models of induced colitis [65,66,67,68,83]. The oral administration or application of enemas containing these drugs, alone or associated with other substances with anti-inflammatory activity, can reduce the inflammatory process of the colonic mucosa, and oxidative tissue damage as well as improve epithelial healing of the colonic mucosa [68,83].

Yuan et al. were the first authors to evaluate the effects of Glycyrrhizinate extract in an experimental model of acetic acid-induced colitis [65]. The authors described that Glycyrrhizinate extract has a potent anti-inflammatory effect that is mediated by the suppression of NF-κB, TNF-α, and ICAM-1 in colonic mucosa. Three years later, Sun et al. investigated the therapeutic potential of G in trinitrobenzene sulfonic acid (TNBS)-induced experimental colitis in mice [66]. After colitis induction by TNBS, G was administered by gavage (10 mg/kg, 20 mg/kg, and 30 mg/kg) for 10 days. G significantly ameliorated TNBS-induced colitis and dose-dependently decreased macroscopic and microscopic inflammation scores, and MPO activity. Mechanistically, G downregulated the colonic levels of the pro-inflammatory cytokines IFN-α, IL-12, TNF-α, and IL-17 and increased the anti-inflammatory cytokine IL-10. The efficacy of G topical application in the treatment of a rat model of UC was also evaluated in experimental colitis models induced by dextran sodium sulfate (DSS). G significantly ameliorated the extent of colitis, which was associated with a decrease in the expression levels of pro-inflammatory cytokines and chemokines, including interleukin IL-1β, IL-6, TNF-α, CXCL2, and CCL2 in the inflamed mucosa. G also inhibited MPO activity in the inflamed mucosa and had a therapeutic effect on experimental colitis in rats [67]. A synergistic effect was observed when G was combined with emu oil in a colitis rat model induced by acid acetic. The authors observed that the treatment combination significantly improved their ability to reduce macroscopic and microscopic lesions as well as to decrease MPO levels and enhanced downmodulation on PPARγ and TNF-α expression [69]. More recently, Chen et al. reported that G ameliorated colitis and decreased the production of inflammatory mediators such as HMGB1, IFN-γ, IL-6, TNF-α, and IL-17. Furthermore, G suppressed the proliferation of Th17 cells in colitis and inhibited the ability of dendritic cells and macrophages to induce the differentiation of Th17 cells that was enhanced in the presence of HMGB1 [70].

To find the best route of administration, Liu et al. compared rectal and oral treatments with GA in TNBS-induced colitis in rats. Both rectally and orally administered treatment effectively attenuated colitis at different dosages. Furthermore, administration by both routes decreased serum levels of TNF-α and IL-1β, colon MPO activity and MDA concentration, and elevated SOD activity [68]. It has been demonstrated that GA is capable of blocking prostaglandin-E2 synthesis via blockade of COX-2 resulting in concurrent augmentation of nitric oxide production on indomethacin-induced small intestinal injury in mice [71]. In addition, using an ulcerative colitis mice model, GA reduced IL-6 and IL-1β, regulating the phosphorylation of NF-κB and IkB-α, and the expression of COX-2 and PGE2 in an ulcerative colitis model [72]. The anti-inflammatory mechanisms of both G and GA were described as mediated by IFN-y, TNF-alpha, IL-4, IL-5, IL-6, IL-8, IL-10, IL-12, and IL-17. Moreover, G and GA mediated intercellular adhesion molecules 1, P-selectin, iNOS, and the NF-κB pathway through the nuclear translocation of NF-κB and activation of STATs 3 and 6 [11]. In human colonic epithelial cell line HT-29, GA exhibits the inhibitory activity on TNF-α and IL-8 production and the blockade of the MAPK and the IKB/NF-κB pathways [63].

The use of DPG as a therapeutic strategy to overcome intestinal inflammation was also evaluated. Vitali et al. studied the DPG effects on HMGB1, an early pro-inflammatory cytokine that is released from injured cells during inflammation. In vitro assays show that DPG significantly reduces the release of HMGB1 as well as expression levels of pro-inflammatory cytokines, TNF-α, IL-1β and IL-6. In vivo, DPG decreases the severity of DSS-induced colitis as well as intestinal inflammation reduction mediated by a downregulation of the pro-inflammatory cytokines TNF-α, IL-1β, and IL-6, as well as HMGB1 receptors, RAGE, and TLR4 [34]. Posteriorly, the same research group showed that DPG has a protective effect on colitis and inflammation through the inhibition of oxidative tissue stress [49]. It was observed that DPG can decrease oxidative stress through the inhibition of iNOS and COX-2 ameliorating DSS-induced colitis in mice. It was demonstrated in vitro that DPG decreases inflammation-related oxidative stress, through (i) an earlier ability to promote AMP-activated kinase (AMPK)-phosphorylation and (ii) a later HMGB1-dependent mechanism. Moreover, DPG can also improve colonic inflammation in DSS-induced colitis mice model through downregulation of the pro-inflammatory genes (CXCL1, CXCL3, CXCL5, PTGS2, IL-1β, IL-6, CCL12, and CCL7), and upregulation of genes involved in healing (COL3A1, MMP9, VTN, PLAUR, SERPINE, CSF3, FGF2, FGF7, PLAT, and TIMP1), which contribute to accelerating intestinal mucosa repair in induced colitis [73]. It has recently been shown that DPG increases the expression of the receptors farnesoid-X-receptor (FXR), pregnane-X-receptor (PXR), and G-protein-coupled-receptor (GPCR; TGR5), decreasing the oxidative stress and consequently intestinal/hepatic inflammation in DSS colitis animal model, with a decrease in IL-8 [84]. Figure 2 summarizes the main mechanism of action of G, GA, and DPG.

The evidence from all these experimental studies suggests that the bioactive compounds from Licorice (*Glycyrrhiza glabra*) have anti-inflammatory and antioxidant effects in intestinal disorders through different mechanisms of action. Considering that these molecular features are also important in human intestinal disorders, it is reasonable to assume that Licorice might have similar activity in humans. Therefore, several clinical studies have focused on the pharmacological effects of Licorice on intestinal diseases (Table 2). However, to date, there is no clinical evidence showing the effect of Licorice in patients with IBD.

It has also been shown that Licorice has potentially serious side effects for humans [27]. Clinical studies have shown that the most important side effects of Licorice and *glycyrrhizin* are hypertension and hypokalemic-induced secondary disorders [85]. Biochemical studies indicate that G inhibits 11beta-hydroxysteroid dehydrogenase, the enzyme responsible for inactivating cortisol. As a result, continuous, high-level exposure to GZ compounds can produce hypermineralocorticoid-like effects in both animals and humans [7,27]. Chronic use of Licorice can lead to hypokalemia and hypertension. Some people are more sensitive to licorice exposure [7]. Licorice side effects are increased by hypokalemia, prolonged gastrointestinal transient time, decreased 11-beta-hydroxysteroid dehydrogenase activities, hypertension, and anorexia nervosa [85]. These side-effects are reversible upon withdrawal of Licorice or Glycyrrhizin [27]. It can be assumed from these data that the consumption of Licorice extract products presents no concern for safe use as a supporting drug in patients with IBD. However, multicenter, randomized clinical studies that include a larger number of patients are still necessary to verify the benefits of using Licorice extracts in the treatment of IBD, and as a natural therapeutic strategy to prevent CRC in patients with extensive and active forms of long-term IBD.

## 5. Conclusions and Future Perspectives

The broad involvement of Licorice-derived compounds in intestinal disorders and their potential to overcome these disorders and the mechanism of action is presented in this review. In summary, the evidence from all these experimental studies suggests that the bioactive compounds obtained from Licorice have anti-inflammatory and antioxidant properties that affect anti-intestinal disorders through different mechanisms of action. This provides an interesting background for understanding how G, GA, and DPG compounds act and contributes to the development of natural therapeutic strategies and to the establishment of research models. In addition, more research is needed to determine the mechanism of action in different biological activities. Clinical trials on G, GA, and DPG are also required to validate these pharmacological effects, to establish these compounds as promising pharmaceuticals, and to fill some gaps regarding their safety and toxicological characteristics.

## Figures and Tables

**Figure 1 ijms-23-04121-f001:**
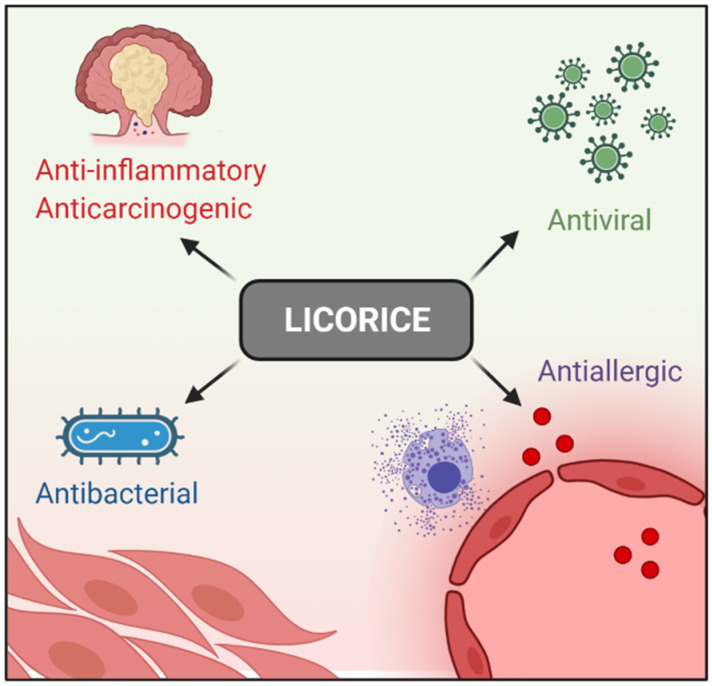
Licorice pharmacological properties.

**Figure 2 ijms-23-04121-f002:**
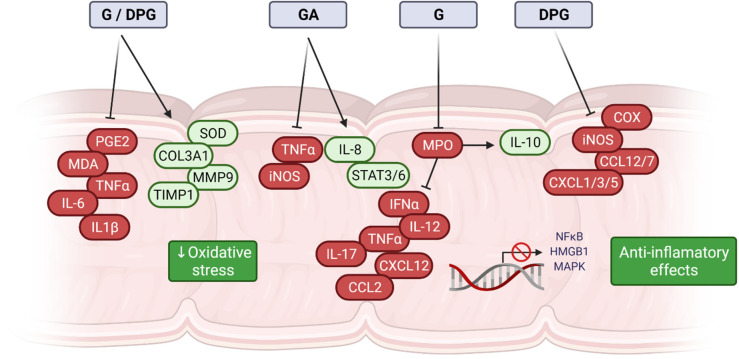
Molecular mechanisms of *Glycyrrhiza glabra*-derived compounds in intestinal disorders. Compounds derived from *Glycyrrhiza glabra* have anti-inflammatory potential. G, GA, and DPG act through the inhibition of HMGB1, TLR4, and RAGE receptors and significantly regulate important cytokines, interleukins, and genes involved in the inflammatory process. These effects are related to the capacity of regulating important inflammatory signaling pathways such as HMGB1, NF-κB, and MAPK. Oxidative stress is significantly reduced because of cellular and molecular changes, and consequently, the inflammatory process is attenuated as a result of treatment with these compounds.

**Table 1 ijms-23-04121-t001:** Summary of studies showing the autoinflammatory and anti-tumoral effects of G, GA, and DPG.

Model	Compound (Dose)	Mechanism	Reference
In vitro (KATO III and HL-60)	G (1 to 10 mg/mL)	Antitumor activity ↑ apoptosis	[23]
In vitro (HLE, KATO III, and HL-60)	G (0.1 to 1 mg/mL)	Antitumor activity ↑ apoptosis	[24]
In vitro (DU-145 and LNCaP)	G (1 to 20 mM)	Antitumor activity ↑ apoptosis	[25]
In vitro (Caco3, HT29, and RAW 264.7) In vivo (Acute lung injury mice model)	DPG (300 µM) DPG (3 and 8 mg/kg/day)	↓ TNF-α, IL-1β, and IL-6, as well as HMGB1 receptors, RAGE and TLR4	[34]
In vitro (neutrophils)	G (0.05, 0.5, and 5.0 µg/mL)	↓ ROS	[38]
In vivo (Con A-induced hepatitis) Ex vivo (liver dendritic cells)	G (2 mg/mouse) G (0.1 mg/mL)	↑ IL-10 and ↓ liver inflammation	[39]
In vitro (U251)	GA (1, 2, 4 mM)	Anticancer effect ↓ proliferation and ↑ apoptosis possibly related to the NF-κB mediated pathway	[40]
In vitro (U87MG and T98G)	DPG (0.1 to 2 mM)	Anticancer effect ↓ proliferation and ↑ apoptosis. ↓ NF-κB pathway	[46]
In vivo (DSS-induced colitis mice model)	DPG (8 mg/kg/day)	↓ colitis, at the earlier stages, ↓ inflammation though AMPK-COX-2-PGE. At later times ↓ iNOS and COX-2 in HMGB1-dependent manner	[49]
In vivo (mechanical thrombectomy rat model)	G (2, 4 and 10 mg/kg/day)	↓ HMGB1 and its downstream inflammatory factors, and ↓ oxidative stress	[50]
In vivo (Focal cerebral I/R injury rat model)	G (4 mg/kg/day)	↓ HMGB1 and ↑ apoptosis through the blockage of the JNK and p38	[51]
In vivo (Sepsis-induced acute lung injury rat model)	G (25 and 50 mg/kg/day)	↓ inflammatory responses, oxidative stress damage, and apoptosis though ↓ NF-κB, JNK, and p38 MAPK	[52]
In vivo (Acute lung injury mice model)	G (20 and 40 mg/kg/day)	↓ LPS-induced lung injury via blocking HMGB1/TLRs/NF-κB pathway	[53]
In vitro (RAW 264.7 and bone marrow monocytes)	G (25 to 100 µM)	↓ RANKL-induced osteoclastogenesis and oxidative stress through ↑ AMPK/Nrf2 and ↓ NF-κB and MAPK	[56]
In vivo (Parkinson rat model)	GA (50 mg/kg/day)	↓ dopamine neuron loss and ↓ Iba-1 and GFAP ↑ antioxidant enzyme activity, ↓ lipid peroxidation, ↓ pro-inflammatory cytokines	[58]
In vivo (Vascular dementia rat model)	GA (20 mg/kg/day)	↓ release of cytochrome-c and ↑ Bcl2, and ↑ the endogenous antioxidants	[59]
In vitro (HBZY-1) In vivo (sepsis-induced acute kidney injury mice model)	GA (50 and 100 µM) GA (25 and 50 mg/kg/day)	↓ oxidative stress via ↑ ERK signaling pathway. ↓ NF-κB	[60]
In vivo (myocardial ischemic injury-rat model)	GA (10 and 20 mg/kg/day)	↓ oxidative stress and inflammatory cytokines. ↑ Nrf2 antioxidant response ↓ NF-κB activation	[61]
In vitro (HEPG2)	G (5, 25 and 125 µg/mL)	↓ H_2_O_2_-induced oxidative stress, ↑ apoptosis	[62]
In vitro (HT29)	GA (1, 5 and 10 µM)	↓ TNF-α-mediated IL-8 through ↓ MAPK and the IKB/NF-κB pathway	[63]
In vivo (DSS-induced colitis mice model)	GA (10 and 50 mg/kg/day)	↓ colitis, ↓ inflammation by regulating COX-2 and NF-κB	[64]
In vivo (rat model of ulcerative colitis)	G (40 mg/kg/day)	↓ colitis, ↓ inflammatory injury via suppression of NF-κB, TNF-α, and ICAM-1	[65]
In vivo (TNBS-induced experimental colitis mice model)	G (10, 30 and 90 mg/kg/day)	↓ colitis, ↓ IFN-γ, IL-12, TNF-α, and IL-17 and ↑ IL-10	[66]
In vivo (DSS-induced colitis rat model)	G (2 mg rectally)	↓ colitis, ↓ IL-1β, IL-6, TNF-α, Cxcl-2, Mcp1, and MPO	[67]
In vivo (TNBS-induced experimental colitis rat model)	GA (2, 10 and 50 mg/kg, rectally and 10 mg/kg/day)	↓ colitis, ↓ serum levels of TNF-α and IL-1β, ↓ colon MPO and MDA, and ↑ SOD	[68]
In vivo (rat model of ulcerative colitis)	G (100 mg/kg/day)	↓ colitis, when combined with emu synergistically ↓ of PPARγ and TNF-α	[69]
In vivo (TNBS-induced experimental colitis mice model)	G (50 mg/kg/day)	↓ colitis, ↓ HMGB1 on DC/macrophage mediated Th17 proliferation	[70]
In vivo (indomethacin-induced small intestinal injury mice model)	GA (100 mg/kg/day)	↓ TNF-α, IL-1β, and IL-6, ↑ indomethacin-induced small intestinal damage	[71]
In vivo (DSS-induced colitis mice model)	G (100 mg/kg/day)	↓ colitis, regulated the phosphorylation of transcription factors such as NF-κB p65 and IκB α	[72]
In vivo (DSS-induced colitis mice model)	DPG (8 mg/kg/day)	↑ mucosal healing by ↓ CXCL1, CXCL3, CXCL5, PTGS2, IL-1β, IL-6, CCL12, CCL7; ↑ wound healing genes COL3A1, MMP9, VTN, PLAUR, SERPINE, CSF3, FGF2, FGF7, PLAT, TIMP1 and ↑ extracellular matrix remodeling genes, VTN, and PLAUR	[73]

**Table 2 ijms-23-04121-t002:** Clinical trials with Licorice in intestinal disorders.

Drug	Clinical Trial	Phase	N of Pts	Status	Diseases	Results
Traditional Chinese Medication (containing 3 g of Licorice)	NCT03135821	2, 3	104	Unknown	Irritable bowel syndrome	NA
Traditional Chinese Medicine (17 g herbal extract containing G)	NCT00676975	2	104	Complete	Irritable bowel syndrome	NA
Modified Gegen Qinlian Decoction (containing 6 g of Licorice)	NCT04057547	1	60	Recruiting	Ulcerative colitis	NA
Modified Gegen Qinlian Decoction (containing 6 g of Licorice)	NCT04312477	1	60	Recruiting	Irritable bowel syndrome	NA
Traditional Chinese Medicine (17 g herbal extract containing 2 g of G)	NCT04368663	NA	100	Recruiting	Pneumatosis cystoides intestinalis	NA

Abbreviation: NA, not available.

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
