# Peer review of "The Anti-Inflammatory Properties of Licorice (Glycyrrhiza glabra)-Derived Compounds in Intestinal Disorders"

_ijms, 2022, doi:10.3390/ijms23084121_

Round 1

Reviewer 1 Report

In the present review article dos Santos Leite et al discussed about anti-inflammatory and anti-oxidant properties of licorice for gastrointestinal disorders.

My only relevant criticism is that all studies that have been analyzed have been performed in vitro on cell cultures or on animal models of colitis. Such models do not mirror perfectly human beings and, furthermore, it seems that there is no evidence about studies or clinical trials on patients with IBD. Based on such observation, there is no solid evidence recommending licorice, nor proof that licorice is really effective on IBD patients. authors should discuss this point, being more cautious in the conclusions.

Minor points. Page 1, I suggest to delete the line about antiviral effect for SARS-CoV2.

Author Response

Comment 1) My only relevant criticism is that all studies that have been analyzed have been performed in vitro on cell cultures or on animal models of colitis. Such models do not mirror perfectly human beings and, furthermore, it seems that there is no evidence about studies or clinical trials on patients with IBD. Based on such observation, there is no solid evidence recommending licorice, nor proof that licorice is really effective on IBD patients. authors should discuss this point, being more cautious in the conclusions.

Answer: We are grateful for the reviewer’s comments. Following his/her recommendation, a new Table was included (Table 2. Clinical trials with Licorice in intestinal disorders). Additionally, the following statement was also added:

Lines 292-297

" Considering that these molecular features are also important in human intestinal disorders, it is reasonable to assume that licorice might have similar activity in humans. Therefore, several clinical studies have focused on the pharmacological effects of licorice on intestinal diseases (Table 2). However, up to date, there is no clinical evidence showing the effect of licorice in patients with IBD. " 

Comment 2)  Minor points. Page 1, I suggest deleting the line about antiviral effect for SARS-CoV2.

Answer: We are grateful for the reviewer’s comments. Following his/her recommendation, such information was removed from the manuscript. 

Reviewer 2 Report

  1. the manuscript needs extensive revision with addition of comprehensive table.
  2. the mechanism needs to be added for glycyrhiza glabra. 
  3. Section 2: the information is inappropriate. the sentences are poorly framed. this section does not matches with title of manuscript.
  4. The information given must added with dose of drug.
  5. Conclusion and future prospect must be elabrated.

Author Response

Comment 1) The manuscript needs extensive revision with the addition of a comprehensive table.

Answer: We are grateful for the reviewer’s comments. Following his/her recommendation, an extensive revision was made. We also added a comprehensive Table (Table 1. Summary of studies showing the autoinflammatory and anti-tumoral effects of G, GA, and DPG).

Comment 2) the mechanism needs to be added for glycyrhiza glabra. 

Answer:  We are grateful for the reviewer’s comments. Following his/her recommendation, apart from Figure 2 (Molecular mechanisms of Glycyrrhiza glabra-derived compounds in intestinal disorders)  we added a new Fig (Figure 1. Licorice pharmacological properties). 

Comment 3) Section 2: the information is inappropriate. the sentences are poorly framed, and this section does not match with the title of the manuscript.

Answer:  We are grateful for the reviewer’s comments. Following his/her recommendation, Section 2 was removed.

Comment 4) The information given must be added with the dose of the drug.

Answer:  We are grateful for the reviewer’s comments. Following his/her recommendation, this information was added in the new Table 1 (Summary of studies showing the autoinflammatory and anti-tumoral effects of G, GA, and DPG).

Comment 5) Conclusion and a future prospect must be elaborated.

Answer:  We are grateful for the reviewer’s comments. Following his/her recommendation, the " Conclusion and a future prospect" was re-written as follow

Lines 322-332

The broad involvement of Licorice-derived compounds in intestinal disorders and their potential to overcome them and the mechanism of action is presented in this review. In summary, the evidence from all these experimental studies suggests that the bioactive compounds obtained from Licorice have anti-inflammatory, antioxidant, and anti- intestinal disorders effects through different mechanisms of action. This provides an interesting background for understanding how G, GA, and DPG compounds act, also contributing to the development of naturally based therapeutic strategies and to the establishment of research models. In addition, more research is needed to determine the mechanism of action in different biological activities. Clinical trials are also required on G, GA, and DPG to validate these pharmacological effects, to establish these compounds as promising pharmaceuticals, and to fill some gaps in their safety and toxicological characteristics.

Round 2

Reviewer 1 Report

Answers were satisfactory

Reviewer 2 Report

Accept